



# Impact of freshwater runoff from the southwest Greenland Ice Sheet on fjord productivity since the late 19th century

Mimmi Oksman[1], Anna Bang Kvorning[1], Signe Hillerup Larsen[1], Kristian Kjellerup Kjeldsen[1], Kenneth David Mankoff[1], William Colgan[1], Thorbjørn Joest Andersen[2], Niels Nørgaard-Pedersen[3], Marit-Solveig Seidenkrantz[4], Naja Mikkelsen[1], Sofia Ribeiro[1]

[1]Department of Glaciology and Climate, Geological Survey of Denmark and Greenland, Øster Voldgade 10, 1350 Copenhagen, Denmark.
[2]Department of Geosciences and Natural Resource Management, University of Copenhagen, Øster Voldgade 10, 1350 Copenhagen, Denmark.
[3]Department of Marine Geology, Geological Survey of Denmark and Greenland, C.F. Møllers Allé 8, 8000 Aarhus C, Denmark.
[4]Paleoceanography and Paleoclimate Group, Arctic Research Centre, and iClimate centre, Department of Geoscience, Aarhus University, Høegh-Guldbergs Gade 2, 8000 Aarhus C, Denmark.

*Correspondence to:* Mimmi Oksman (mio@geus.dk)

**Abstract.** Climate warming and the resulting acceleration of freshwater discharge from the Greenland Ice Sheet are impacting Arctic marine coastal ecosystems, with implications for their biological productivity. To accurately project the future of coastal ecosystems, and place recent trends into perspective, paleo-records are essential. Here, we present late 19[th] century to present runoff estimates for a large sub-Arctic fjord system (Nuup Kangerlua, southwest Greenland) influenced by both marine- and land-terminating glaciers. We followed a multiproxy approach to reconstruct spatial and temporal trends in primary production from four sediment cores, including diatom fluxes and assemblage composition changes, biogeochemical and sedimentological proxies (total organic carbon, nitrogen, C/N-ratio, biogenic silica, $\delta^{13}C$, $\delta^{15}N$, grain size distribution). We show that an abrupt increase in freshwater runoff in the mid-1990's is reflected by a 3-fold increase in biogenic silica fluxes in the glacier-proximal area of the fjord. In addition to increased productivity, freshwater runoff modulates the diatom assemblages and drives the dynamics and magnitude of the diatom spring bloom. Our records indicate that marine productivity is higher today than it has been at any point since the late 19[th] century and suggest that increased mass loss of the Greenland Ice Sheet is likely to continue promoting high productivity levels at sites proximal to marine-terminating glaciers. We highlight the importance of paleo-records in offering a unique temporal perspective on ice-ocean-ecosystem responses to climate forcing beyond existing remote sensing or monitoring time-series.

## 1 Introduction

Arctic coastal ecosystems are experiencing profound and rapid changes that are predicted to intensify in the near future due to ongoing climate changes (Lannuzel et al., 2020). Arctic ecosystems are impacted by sea-ice thinning and retreat (Arrigo and van Dijken, 2015; Stroeve and Notz, 2018) and by increasing freshwater discharge from melting glaciers and terrestrial runoff (Hopwood et al., 2020). During the last two decades, freshwater runoff from the Greenland Ice Sheet (GrIS) has rapidly increased and doubled in volume, amounting to ca. 1 000 Gt year[-1] today (Bamber et al., 2018; Mankoff et al., 2020), and under a warming climate, freshwater discharge will continue to increase in the future (Fettweis et al., 2013). Glacial freshwater discharge changes the physical and chemical properties of the water column and nutrient


(nitrate, phosphate and silica) dynamics, which in turn impacts primary production, the length of the productive season, species composition and the distribution of biological communities (e.g., Hawkings et al., 2015; Hopwood et al., 2020). Recent studies have observed increased phytoplankton blooms and primary production rates as well as a more prolonged productive season in the high Arctic (Renaut et al., 2018, Lewis et al., 2020). However, these studies cover only the last

few decades whereas long-term records, providing the natural baseline for the recent changes, are scarce.

Greenland fjord ecosystems, at the interface between the ocean and the GrIS, maintain a high biological productivity, which is essential to sustain important socio-economic activities such as fisheries and harvesting of indigenous resources (mainly marine mammals). Higher-trophic-level organisms, such as mammals and birds, are typically not directly impacted by the freshwater increase yet changes in their food sources have cascading effects with severe implications to

these organisms (e.g., Berthelsen, 2014). Long-term records of primary production changes, as well as knowledge on its spatial variability, are needed to estimate the fate of these unique ecosystems under a changing climate. However, our current knowledge of freshwater impacts on marine productivity is largely based on modern observations (Meire et al., 2017; Hopwood et al., 2018, 2020) and on monitoring data spanning only a few decades (Greenland Ecosystem Monitoring program; GEM). These studies have shown that the influence of glaciers on marine productivity depends on

multiple factors such as glacier type (i.e., marine- vs. land-terminating), fjord geometry, and on the resources (nutrients and light availability) that are limiting phytoplankton growth (Hopwood et al., 2020). In fjords with marine-terminating glaciers, sub-glacial discharge plays a key role in sustaining high productivity levels, as the upwelling meltwater plume brings crucial nutrients from the ambient deep water to the surface (Meire et al., 2017; Hopwood et al., 2018; Kanna et al., 2018). However, in the future, this nutrient source can shut down as marine-terminating glaciers retreat onto land or

if glacier grounding line depth changes significantly.

The GrIS is losing mass (between 10s and ~500 Gt yr$^{-1}$) with a long-term trend of increasing mass loss, and this shrinking is most pronounced on the western side of the ice sheet (Mankoff et al., 2021). Nuup Kangerlua (Godthåbsfjord), situated in southwest Greenland, (Fig. 1) is one of the largest fjord systems in the world. The fjord is ca. 190 km long, 4 – 8 km wide, up to 625 m deep, and covers an area of ca. 2 013 km$^2$. The fjord has several sills; the main one is located at the

mouth of the fjord at ca. 170 m water depth, with smaller sills in the side branches of the fjord (Mortensen et al., 2011). The fjord hydrography is influenced by ocean forcing (governed largely by the West Greenland Current, WGC) and both tidal and wind mixing (Juul-Pedersen et al., 2015), and the combination of atmosphere-ocean-ice forcing makes the fjord particularly sensitive to climatic changes. The main fjord branch, Nuup Kangerlua, is heavily influenced by glacial discharge and functions as a gateway for iceberg outflow, while the three side branches Kapisillit Kangerluat, Qoornup

Sullua and Uummannap Sullua, do not house large outlet glaciers and are not directly influenced by glacial discharge (Fig. 1). The average yearly freshwater discharge between 2015 and 2019 was ca. 20.9 Gt year$^{-1}$ and solid ice discharge ca. 8.1 Gt year$^{-1}$ (Mankoff et al., 2020). The fjord system is heavily impacted by meltwater, especially during summer, which lowers the surface salinities (Meire et al., 2017). However, icebergs also have a significant role in the freshwater flux as they can produce up to 22 % of the total meltwater (Van As et al., 2014). The fjord receives freshwater and

sediment discharge from three marine-terminating glaciers (Kangiata Nunaata Sermia, KNS; Akullersuup Sermia, AS; and Narsap Sermia, NS) and three land-terminating glaciers (Qamanaarsuup Sermia; QS, Kangilinnguata Sermia; KS and Saqqap Sermia; SS) of which SS drains into Lake Tasersuaq, while the others drain into the fjord (Fig. 1). A large interannual environmental gradient exists inside the fjord as the inner part receives ice, freshwater and sediments especially in the spring and late summer, creating salinity, temperature, and productivity gradients (Mortensen et al.,



2013; Krawczyk et al. 2015a, 2018; Meire et al., 2016a, 2017). The turbid meltwater reduces light penetration in the inner fjord close to the outlet point of Lake Tasersuaq creating a large spatial contrast across the fjord (Fig. 1; Meire et al., 2017). The inner part of the fjord is annually covered by sea ice from November to May, whereas the outer part stays ice free for most of the year.

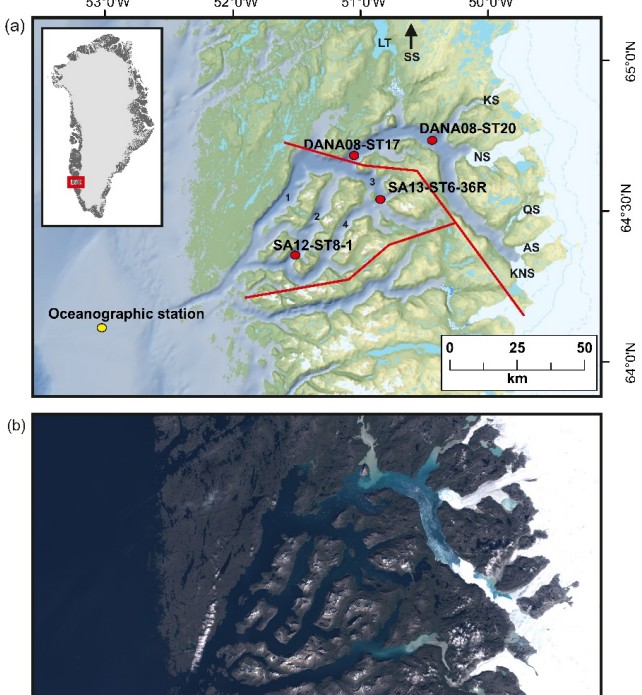

**Figure 1.** (a) Locations of studied sediment cores SA13-ST6-36R, SA12-ST8-1, DANA08-ST17, and DANA08-ST20, and the main
freshwater sources feeding into the fjord; Kangiata Nunaata Sermia (KNS), Akullersuup Sermia (AS), Narssap Sermia (NS), Kangilinnguata Sermia (KS), Qamanaarsuup Sermia (QS), Saqqap Sermia (SS) and Lake Tasersuaq (LT) in the Nuup Kangerlua fjord system. 1) Main fjord branch Nuup Kangerlua, 2) Qoornup Sullua, 3) Kapisillit Kangerluat, and 4) Uummannap Sullua. Yellow circle indicates the oceanographic station for sea-surface temperature measurements (Ribergaard et al., 2014). Redlines mark the division
into inner, outer, and southern fjord used for the freshwater runoff estimates. (b) True color Sentinel-2 imagery of the fjord obtained from http://earthexplorer.usgs.gov (last access: 30 Nov 2021).

Present-day total annual primary production in Nuup Kangerlua is generally high (84.6 – 139.1 g C m$^{-2}$ yr$^{-1}$) and it is maintained by diverse pelagic and benthic protist communities (Juul-Pedersen et al., 2015; Krawczyk et al., 2015a,
Krawczyk et al., 2015b, 2018). Two phytoplankton blooms, of a similar magnitude, occur each year with the first bloom taking place during the spring in April/May and the second during the summer in July/August (Juul-Pedersen et al., 2015). The spring bloom plays a major role in Nuup Kangerlua accounting for 50 – 60 % of the annual production, and it is highly relevant for fueling the marine food web by starting the annual productive season (Hodal et al., 2012, Meire et al., 2016a). During recent decades, phytoplankton spring blooms in the Arctic have become markedly longer and they occur



earlier due to sea-ice retreat (Kahru et al., 2011). However, in Arctic fjords, increased freshwater discharge has been suggested to be one of the main factors enhancing the phytoplankton spring bloom (Meire et al., 2016a). In Nuup Kangerlua, the timing, intensity, and location of the modern spring bloom is also controlled by a combination of sea-ice cover, upwelling of nutrient-rich waters, and wind forcing (Juul-Pedersen et al., 2015; Meire et al., 2015, 2016a). The spring bloom is typically dominated by pennate diatoms (that form silicified frustules) and thus the input of dissolved

silica (DSi) from the GrIS is one of the critical components driving the primary production. Meltwater runoff is suggested to be tightly linked to DSi input to the fjord, and surface runoff contributes 79 % of the total silica input to Nuup Kangerlua, whereas subglacial discharge accounts only for 12 % and solid ice discharge 9 % of the silica input (Meire et al., 2016b). Together, these three mechanisms sustain a high silica concentration and primary production 30 – 80 km down fjord from the marine terminating glaciers, with the highest production at ca. ~50 km off the glacier termini (Meire

et al., 2016b, 2017).

Diatoms are one of the principal contributors to marine primary productivity in polar and sub-polar regions. The silicified frustules of diatoms sink to the seabed and, over time, become archived in sedimentary records (Hasle and Syvertsen, 1996). While diatoms are widely distributed and abundant as a group, many species have narrow ecological preferences in terms of hydrographic conditions (salinity, temperature, and light availability) and this feature makes them an excellent

tool for reconstructing past environmental and productivity changes (Hasle and Syvertsen, 1996; von Quillfeldt, 2000, 2004; Krawczyk et al., 2018; Oksman et al., 2019; Weckström et al., 2020). In Nuup Kangerlua, diatom assemblages have been shown to respond mainly to physical conditions in the water column (temperature and salinity) rather than to biogeochemical properties (e.g., nutrients), and especially glacial discharge favors diatoms mostly when compared to other protists groups (Krawczyk et al., 2015a, 2018).

Recent studies on primary production around Greenland have focused either on process-level understanding, characterizing dynamics during the productive season, or on the annual cycle of species succession (e.g., Arendt et al., 2016; Meire et al., 2016a; Krawczyk et al. 2015a, 2015b, 2018; Holding et al., 2019; Luostarinen et al., 2020). Although some marine and coastal studies have focused on longer timescales (e.g., Limoges et al., 2020; Saini et al., 2020), records of decadal to centennial fjord productivity and particularly its response to increased freshwater discharge are poorly

documented. As underlined by the IPCC SROCC 2019 report, long-term variations in fjord primary productivity remain poorly constrained, and thus we lack a solid knowledge of past productivity responses to cryosphere changes that can be used as a base for future projections (Bindoff et al., 2019).

The aim of this study is to assess the impacts of freshwater runoff from the GrIS on primary productivity in the Nuup Kangerlua fjord system since the late 19th century (from ca. 1890 to present day). We chose Nuup Kangerlua as a suitable

glaciated fjord system due to its high productivity, magnitude of modern glacial discharge, and because the fjord has been monitored since 2008 and is a key site for the GEM. We extracted basin-wide freshwater runoff estimates between 1900 and 2019 from model hindcasts and reconstructed trends in primary production from four dated sediment records following a multiproxy approach including diatom fluxes and assemblage composition, biogenic silica (BSi) content, composition and origin of organic matter (TOC, C/N-ratio, carbon and nitrogen isotopes; $\delta^{13}C$ and $\delta^{15}N$), and grain-size

analysis. Alongside with the freshwater runoff estimates, we interpret spatial and temporal trends in our proxy data against historical records of regional air and sea surface temperature, and changes in the front position of glaciers feeding into the fjord. Our combined records of freshwater runoff and primary production span more than a century and provide a novel insight into historical and recent changes and their spatial variability.



## 2 Material and methods

### 2.1 Sediment core material

A total of four sediment core records were included in this study, retrieved during three campaigns that took place between 2008 and 2013 (Fig. 1). During the campaigns, coring sites were carefully selected based on shallow seismic records and multibeam bathymetry data from previous research cruises (Harff et al., 2016; Jensen and Rasch, 2009; Mikkelsen et al., 2012, Seidenkrantz et al., 2014). The studied sediment cores were chosen based on their location and temporal resolution.

Core SA12-ST8-1 (503 m water depth, 153 cm long, 64°16,416`N 51°19,777`W), was retrieved from the side branch Qoornup Sullua in May 2012 (Mikkelsen et al., 2012) and core SA13-ST6-36R (398 m water-depth, 96 cm-long, 64°29,076`N 50°42,553`W) was retrieved from the side branch Kapisillit Kangerluat in 2013 (Seidenkrantz et al. 2013). Both cores were collected on board the Greenlandic RV *Sanna* using a Rumorh-lot corer. Cores DANA08-17 (575m water-depth, 40 cm-long, 64°37,277`N 50°56,938`W) and DANA08-20 (496 m water-depth, 45 cm-long, 64°41,927`N

50° 21,566`W) were recovered with a multicorer from the main branch of Nuup Kangerlua in 2008, on board the Danish RV *Dana*. This fjord branch is heavily influenced by glacial discharge, as ice floes and icebergs travel through this channel to exit the fjord (Fig. 1). All sediment cores were stored in cold (4 – 7 °C) and dark conditions until opened. Sediment core sub-samples were frozen at -20 °C and freeze-dried before further processing. Sediment core records are hereafter referred to by their site numbers as site 8 (SA12-ST8-1), 6 (SA13-ST6-36R), 17 (DANA08-17) and 20 (DANA08-20)

and presented in the figures following an outer-to-inner fjord gradient (8, 6, 17, and 20).

### 2.2 Geochronology

The activity $^{210}$Pb, $^{226}$Ra and $^{137}$Cs was analyzed from all studied cores by gamma-spectrometry at the Gamma Dating Center at the University of Copenhagen. From selected depth intervals, ca. 2 gram of freeze-dried sediment was measured using Canberra ultralow-background Ge-detectors. $^{210}$Pb was measured via its gamma-peak at 46.5 keV, $^{226}$Ra via the

granddaughter $^{214}$Pb (peaks at 295 and 352 keV) and $^{137}$Cs via its peak at 661 keV. Chronologies for each site were obtained using a modified constant rate of supply (CRS) model (Andersen, 2017), where the activity below the lowermost sample was calculated based on a regression of activity versus accumulated mass depth for the upper part of the core. The obtained chronologies were validated by $^{137}$Cs data.

### 2.3 Grain size analysis

The grain-size distribution of the sediment records was analyzed by wet sieving a minimum of 2 grams of freeze-dried sediment per sample, in order to obtain the percentages of clay and silt grains (<63 μm), fine sand (63 – 150 μm), and coarse sand (>150 μm). Further analysis, using laser diffraction with a Malvern Mastersizer E/2000, was done to obtain the distribution of clay (<2 μm) and silt (2 – 63 μm) grains. Grain-size analysis was measured from 28, 19, 20, and 23 samples from sites 8, 6, 17, and 20 (respectively). Before measurements, all samples were treated with a sodium

pyrophosphate decahydrate-solution and ultrasonicated once for 2 minutes to separate aggregates.

### 2.4 Biogeochemical analysis

The total organic carbon (TOC) content of the sediments was measured at the Geological Survey of Denmark and Greenland using a LECO CS-200 instrument on sub-samples (1 g dried weight). TOC was measured in a total of 72





samples: 19 from site 8, 10 from site 6, 20 from site 17, and 23 from site 20. The inorganic carbon fraction was removed by treating the samples with HCl before measuring their TOC content. Sources of the organic material (terrestrial versus marine) were further investigated from 13 (site 8) and 10 (site 6) samples by measuring the isotopic composition of bulk organic carbon ($\delta^{13}C$) and nitrogen ($\delta^{15}N$). The isotopic composition analyses were done at the IRMS lab, Department of Geosciences and Natural Resources Management, University of Copenhagen. Ca. 20 – 30 mg per sample were homogenized before their carbon and nitrogen content and isotopic ratios were measured by Dumas combustion (1700 °C) on an elemental analyzer (CE 1110, Thermo Electron, Milan, Italy or Flash 2000, Thermo Scientific, Bremen, Germany). The EA is coupled in continuous flow mode to a Finnigan MAT Delta PLUS or Thermo Delta V Advantage isotope ratio mass spectrometer (Thermo Scientific, Bremen, Germany).

Biogenic silica was measured at the Geological Survey of Denmark and Greenland according to the 5h $Na_2CO_3$ extraction method (DeMaster, 1991) in a total of 100 samples; 40, 17, 20 and 23 samples from sites 8, 6, 17, and 20, respectively. Samples of $30 \pm 1$ mg of freeze-dried sediment were leached using 40 ml of 1 % sodium carbonate ($Na_2CO_3$) in an 85 °C water bath for 5 hours. Subsamples (1 ml) were taken after 3, 4, and 5 hours of extraction and measured with a spectrophotometer Jenway 7305 using the blue ammonium-molybdate method (Mullin and Riley, 1955). Final biogenic silica concentrations were calculated using linear regression under the assumption that all biogenic silica dissolved during the first two hours, whereas minerogenic silica dissolves slowly at a constant rate throughout the extraction (Barão et al. 2015).

**2.5 Diatoms**

Diatom microscopy slides were prepared according to Battarbee et al., (2001) at the Geological Survey of Denmark and Greenland from 48, 21, and 20 samples from sites 8, 6, and 17, respectively. Ca. 0,1 grams of sediment per sample was treated with 10 % hydrochloric acid (HCl) and 30 % hydrogen peroxide ($H_2O_2$) in 80 °C water bath for 4 hours to remove carbonates and organic material. Afterwards, samples were rinsed with distilled water and spiked with 0.1 ml of a microsphere solution of 6.13 x $10^6$ spheres/ml. Slides were then mounted using Naphrax® with a refractive index of 1.73. Diatom slides were analyzed with an Olympus BX51 microscope using phase contrast optics at 1000X magnification, and a minimum of 300 valves were counted (excluding *Chaetoceros*). Diatom concentrations (valves $g^{-1}$ dry sediment) were calculated using the sum of microspheres counted per total diatom sum and the number of microspheres added in the diatom sample. The non-linear ordination method of correspondence analysis (CA) was utilized using the program PAST version 4.02 (Hammer et al., 2001) to detect ecological shifts and the major patterns of variation in the diatom assemblages. CA was run on relative abundance data for all taxa representing >1 %, and the resulting sample eigenvalues give a measure of the variance accounted for by the corresponding eigenvectors. To test correlation between CA axis scores, diatom taxa, grain size, geochemical parameters, and environmental variables (runoff, air and sea surface temperatures) a Pearson product-moment correlation coefficient and the associated level of statistical significance ($p <$ 0.05) were used on the 5-year running mean average for each time series.

**2.6 Freshwater runoff 1900-2019**

We used regional climate model data to obtain runoff estimates for the period 1900-2019. For the period 1979-2019, we used the Greenland liquid water runoff dataset (Mankoff et al., 2020). This dataset is based on MAR v3.11, which is driven by the most recent reanalysis data ERA5 (Delahasse et al., 2020). For the period prior to 1979, we use MAR v3.5



(Fettweis et al., 2017) that is based on 20CRv2c reanalysis (Compo et al., 2011). The runoff data from the Greenland liquid water discharge dataset is considered to be more accurate than the MAR v3.5 dataset and thus, the latter data series was calibrated prior to merging of the two datasets. Modelled runoff from both ice and tundra surfaces were extracted for the Nuup Kangerlua basin partitioned into three regions (Fig. 1); inner, outer, and southern part of the fjord in order to
investigate spatial differences between the four sediment core sites.

**2.7 Glacier front position**

To assess temporal changes in front positions of the outlet glaciers feeding into Nuup Kangerlua, we use historical aerial and satellite imagery. The historic imagery consists of oblique and vertical images from the Department of Data-supply and Efficiency, supplemented with Corona-imagery from the 1960s and satellite data from different Landsat missions
retrieved from http://earthexplorer.usgs.gov (last access: 30 Nov 2021). All imagery was georeferenced relative to a 2 m ortho-photo mosaic (Korsgaard et al., 2016), which in this area of Greenland is based on aerial stereophotogrammetric imagery from summer 1985. The georeferencing is based on the method outlined by Bjørk et al., (2012).

**3. Results**

**3.1 Geochronology**

The relatively high sedimentation rates in the fjord since the late 19th century allowed for obtaining records with a sub-decadal resolution for all the cores (average sample resolution between 2 and 6 years).

For site 8, the $^{210}$Pb-based chronology indicates that the upper 55 cm of the record cover the time between ca. 1890 and 2012 (Fig. 2), resulting in an average sample resolution of 2 years. Sedimentation rates are relatively high at site 8 and vary between 0.33 and 0.67 cm yr$^{-1}$, mean 0.45 ± 0.1 cm yr$^{-1}$ (Fig. 3).
For site 6, the topmost 21 cm of the sediment core used in this study represent the period from ca. 1890 to 2013, thus giving an average sample resolution of ca. 6 years (Fig. 2). Sedimentation rates are thus lower at site 6 than site 8 and vary between 0.13 and 0.25 cm yr$^{-1}$, mean 0.18 ± 0.03 cm yr$^{-1}$ (Fig. 3).

The cores from sites 17 and 20 cover shorter time intervals, from 1930 to 2008 and from 1956 to 2008, respectively (Fig. 2). Sedimentation rates are higher in these glacier-proximal sites and vary between 0.24 and 1.18 cm yr$^{-1}$ at site 17 (mean
0.57 ± 0.2 cm yr$^{-1}$) and between 0.46 and 2.33 cm yr$^{-1}$ (mean 1.10 ± 0.6 cm yr$^{-1}$) at site 20 (Fig. 3). The sediment cores from sites 17 and 20 were analyzed at 4-year and 2-year sample resolutions, respectively.

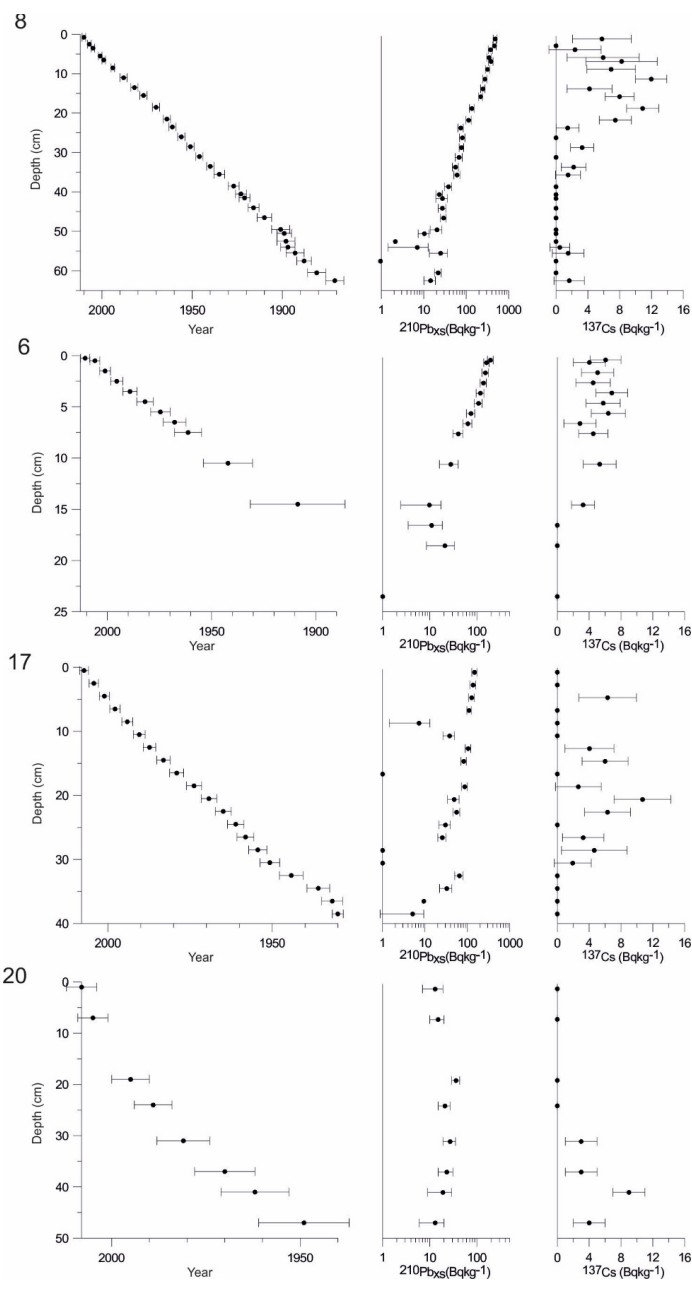

**Figure 2.** Age-depth models, and $^{210}$Pb and $^{137}$Cs activities for the sediment cores SA12-ST8-1 (8), SA13-ST6-36R (6), DANA08-17 (17) and DANA08-20 (20).



## 3.2 Grain-size distribution analysis

All the sediment cores consisted mostly of fine (<63 µm) clay and silt grains (Fig. 3). The <63 µm fraction varied from 90.7 to 98.2 % (mean 96.6 ± 1.4 %) at site 8, from 28.2 to 93.0 % at site 6 (mean 76.7 ± 14.7 %), from 87.3 to 99.0 % (mean 94.2 ± 3.2 %) at site 17, and from 88.5 to 99.6 % (mean 96.7 ± 2.8 %) at site 20 (Fig. 3). More detailed distribution between silt (2 – 63 µm) and clay (<2 µm) grains is presented in Figure 3. The fraction of coarse material >63 µm is widely used as a proxy for ice rafted debris (IRD) carried by icebergs that generally melt relatively close to their source. The fine sand fraction (63 – 150 µm) varied between 1.2 and 3.7 % (mean 2.2 ± 0.7 %) at site 8, between 3.8 and 25.5 % (mean 7.5 ± 4.6 %) at site 6, between 0.6 and 4.8 % (mean 2.1 ± 1.1 %) at site 17, and between 0.2 and 10.8 % (mean 1.9 ± 2.4 %) at site 20, whereas the portion of coarse sand fraction (>150 µm) ranged between 0.2 and 5.8 % (mean 1.2 ± 1.0 %) at site 8, between 4.2 and 22.2 % (mean 8.4 ± 4.4 %) at site 6, between 0.5 and 10.9 % (mean 3.7 ± 2.4 %) at site 17, and between 0.1 and 8.2 % (mean 1.4 ± 1.7 %) at site 20 (Fig. 3).

## 3.3 Biogeochemical analysis

Total organic carbon (TOC) content in sediments provides an integrated signal of production, deposition, and preservation of organic matter over time, whereas the stable carbon and nitrogen isotopes and C/N-ratio can be used to infer the origin of organic matter (i.e., sourced from the terrestrial, marine, or sea ice environments). TOC contents were highest in the most glacial-distal area (sites 8 and 6) and substantially lower at the glacial-proximal sites (17 and 20) (Table 1 and Fig. 3). Sediment C/N-ratios were calculated from the sediment cores from sites 8 and 6 and show relatively low values suggesting a predominantly marine origin (Table 1 and Fig. 3).

**Table 1.** Results of the TOC, TN, C/N-ratio, $\delta^{13}C$, $\delta^{15}N$, and BSi analyses.

| Site | TOC | TN | C/N-ratio | $\delta^{13}C$ | $\delta^{15}N$ | BSi | BSi flux |
|---|---|---|---|---|---|---|---|
| 8 | 1.25 – 1.68 wt% (mean 1.4 ± 0.1 wt%) | 0.28 – 0.33 % (mean 0.3 ± 0.02 %) | 4.6 – 8 (mean 6.3 ± 1.0) | -23.3 – -21.4 ‰ (mean -22.4 ± 0.6 ‰) | 3.9 – 4.7 ‰ (mean 4.3 ± 0.3 ‰) | 21.1 – 35.4 mg/g$^{-1}$ (mean 27.5 ± 3.7 mg/g$^{-1}$) | 7.1 – 19.2 mg cm$^2$ yr$^{-1}$ (mean 12.1 ± 2.9 mg cm$^2$ yr$^{-1}$) |
| 6 | 0.72 – 1.08 wt% (mean 0.7 ± 0.08 wt%) | 0.10 – 0.14 % (mean 0.1 ± 0.01 %) | 6.6 – 8.2 (mean 7.5 ± 0.4) | -23.6 – -22.7 ‰ (mean -23.1 ± 0.3 ‰) | 4.9 – 6.2 ‰ (mean 5.9 ± 0.3 ‰) | 12.8 – 34.5 mg/g$^{-1}$ (mean 22.2 ± 6.0 mg/g$^{-1}$) | 2.1 – 5.8 mg cm$^2$ yr$^{-1}$ (mean 3.9 ± 1.1 mg cm$^2$ yr$^{-1}$) |
| 17 | 0.09 – 0.30 wt% (mean 0.2 ± 0.06 wt%) | | | | | 5.3 – 28.9 mg/g$^{-1}$ (mean 12.6 ± 6.6 mg/g$^{-1}$) | 2.2 – 18.0 mg cm$^2$ yr$^{-1}$ (mean 7.5 ± 5.1 mg cm$^2$ yr$^{-1}$) |
| 20 | 0.007 – 0.07 wt% (mean 0.03 ± 0.02 wt%) | | | | | 5.3 – 14.3 mg/g$^{-1}$ (mean 8.0 ± 2.4 mg/g$^{-1}$) | 3.0 – 25.6 mg cm$^2$ yr$^{-1}$ (mean 9.2 ± 6.3 mg cm$^2$ yr$^{-1}$) |

The stable nitrogen isotope composition in sediments can be used to infer nitrate (NO$_3^-$) supply in the euphotic zone, as phytoplankton consumes lighter nitrate the sediment becomes enriched in the heavier ($\delta^{15}N$) nitrate indicating of increased productivity. The total nitrogen and isotopic composition of the sediments were not measured in the sediment cores from sites 17 and 20 due to the very low organic matter content (Table 1 and Fig. 3). Biogenic silica was highest in the glacial-distal sites than glacier-proximal sites (Table 1) yet changes in sedimentation rates were considered and BSi fluxes were higher in the glacial-proximal sites (Table 1 and Fig. 3).

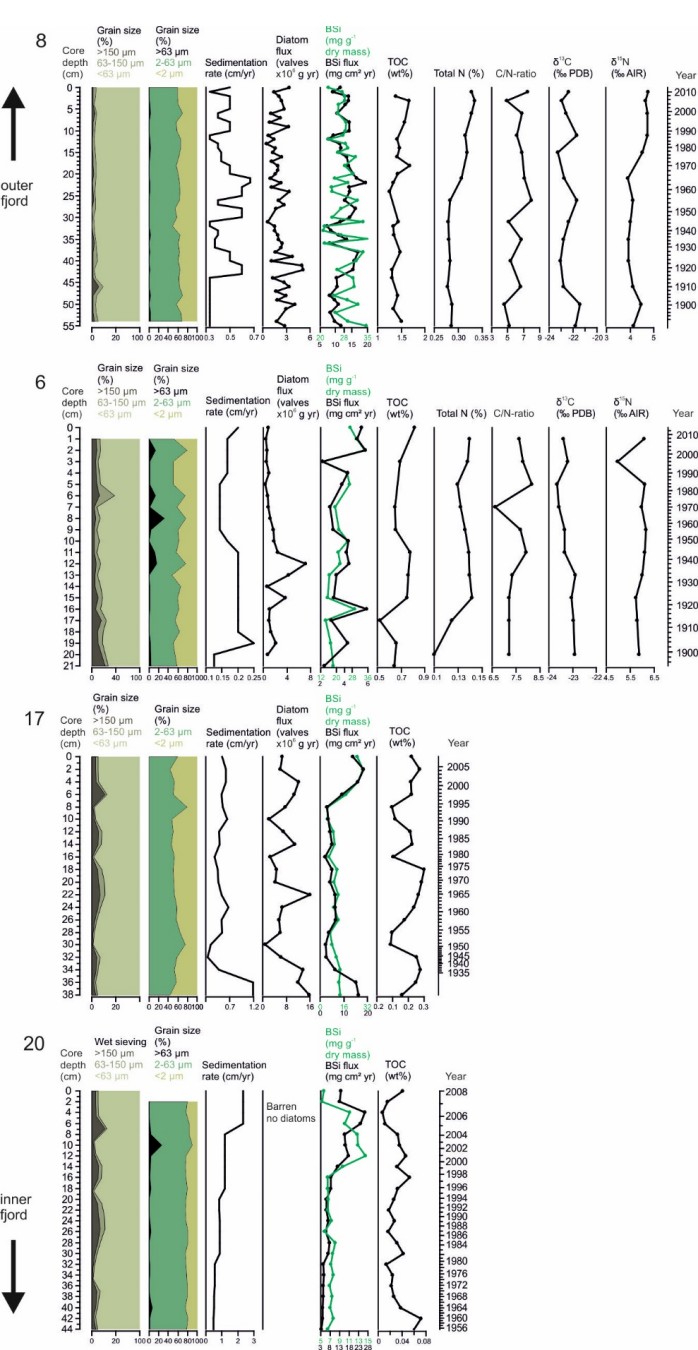

**Figure 3.** Sediment grain-size distribution (wet-sieving and Malvern), sedimentation rate (cm/yr), diatom flux (valves x $10^6$ g yr$^{-1}$), BSi content (mg g$^{-1}$ dry mass); green line and BSi flux (mg cm$^2$ yr$^{-1}$); black line, total organic carbon (wt%), total nitrogen (%), C/N-



270    ratio, $\delta^{13}C$ (‰PDB) and $\delta^{15}N$ (‰AIR) in sediment cores SA12-ST8-1 (8), SA13-ST6-36R (6), DANA08-17 (17) and DANA08-20
         (20).

**3.4 Diatoms**

Diatom assemblages were both rich and diverse at all sites expect site 20, which was barren of diatoms and is discussed
in Section 4.1. At sites 8, 6, and 17, a total of 172, 138, and 93 different taxa were identified to species level and 22, 21,

275    and 10 taxa to genus level, respectively. Total diatom concentrations (including resting spores and vegetative forms of
*Chaetoceros* spp.)in the sediment samples varied from 1.2 to 12.2 x $10^6$ valves/g at site 8, from 2.3 to 35.8 x $10^6$ valves/g
at site 6, and from 2.8 to 30.4 x $10^6$ valves/g at site 17 (Fig. 4). Annual diatom fluxes ranged between 0.5 and 5.1 x$10^6$
valves cm$^{-1}$yr$^{-1}$ (mean 2.1 ± 1.0 x $10^6$ valves cm$^{-1}$yr$^{-1}$) at site 8, 0.3 to 7.2 x$10^6$ valves cm$^{-1}$yr$^{-1}$ (mean 1.6 ± 1.6 x$10^6$ valves
cm$^{-1}$yr$^{-1}$) at site 6 and between 0.8 and 15.8 x$10^6$ valves cm$^{-1}$yr$^{-1}$ (mean 7.8 ± 4.3 x$10^6$ valves cm$^{-1}$yr$^{-1}$) at site 17 (Fig. 3).

280    Diatom assemblage composition was relatively uniform between the three studied sites and diatom taxa (excluding
*Chaetoceros* species) were generally dominated by *Detonula confervacea* (resting spore), *Thalassiosira antarctica* var
*borealis* (resting spore), *Fragilariopsis oceanica*, *F. cylindrus* and *T. nordenskioeldii* (Fig. 4). Diatoms were divided into
6 groups according to their ecological preferences; cold-water species, early spring bloomers, benthic species, sea-ice
associated species and freshwater species (Table 2).


**Table 2.** Diatom groups, species in each group, ecological preferences, and references.

| Group | Species | Ecology | Reference |
|---|---|---|---|
| Cold-water species | *Detonula confervacea*, *Thalassiosira antarctica* var *borealis* (r.s. and veg), *Bacteriosira bathyomphala*, *Thalassiosira anguste-lineata*, *Thalassiosira hyalina*, and *Shionodiscus trifultus* | Species preferring relatively cold waters, blooming in the late summer. | Krawzcyk et al., 2015a; Oksman et al., 2019; Luostarinen et al., 2020 |
| Early spring bloomers | *Fragilariopsis oceanica*, *Fragilariopsis cylindrus*, *Fragilariopsis reginae-jahniae*, and *Pauliella taenitae* | First species to bloom in the spring during and immediately after sea ice break-up and melt. | Quillfeldt 2000, 2004; Oksman et al., 2019; Weckström et al., 2020 |
| Sea-ice associated | *Actinocyclus curvatulus*, *Coscinodiscus oculus-iridis*, *Fossula arctica*, *Porosira glacialis*, *Thalassiosira bulbosa*, *Rhisozolenia* spp., *Pleurosigma stuxbergii* | Species associated with seasonal and high sea ice concentrations or produce sea ice biomarker (HBI III). | Oksman et al., 2019; Weckström et al., 2020, Belt et al., 2017 |
| Freshwater species | *Eunotia* spp., *Cymbella* spp., *Cyclotella* spp., *Tabellaria* spp., and *Pinnularia* spp. | Species living in freshwater environments. | Cremer, 1998; Foged, 1973 |

*Chaetoceros* spp. are generally abundant in the sediment due to their good preservation and are likely to be over-

represented in the assemblage and thus were left out of the diatom assemblage percentage calculations discussed here
         (Fig. 4). However, to have a general idea of the concentrations of *Chaetoceros* spp. and *Chrysophyte* cysts, we calculated
         flux rates (Fig. 4). *Chaetoceros* spp. (including vegetative forms and resting spores) were abundant in all the cores
         containing diatoms, making average flux rates of 0.4, 0.4, and 0.8 x $10^6$ valves cm$^{-1}$yr$^{-1}$ at sites 8, 6, and 17 respectively
         (Fig. 4). Additionally, *Chrysophyte* cysts were counted, as this genus is associated with low-salinity waters during the





summer/autumn bloom (Hasle and Heimdal, 1998; Krawczyk et al., 2015a) and can thus provide additional information
       on past salinity changes.

       For the diatom record from site 8, the first CA axis explains 36 % of the variation in the dataset (Fig. 4). The axis scores
       switch from mainly positive to mostly negative values around the late 1940´s. At site 6, the first CA axis accounts for
       49.7 % of the variation in the dataset and the CA axis scores change from negative to positive at ca. 1960´s. At site 17,

the first CA axis explains 29.5 % of the variation in the dataset. Scores for the first CA axis remain mainly negative until
       ca. 1985 when they shift to positive values.


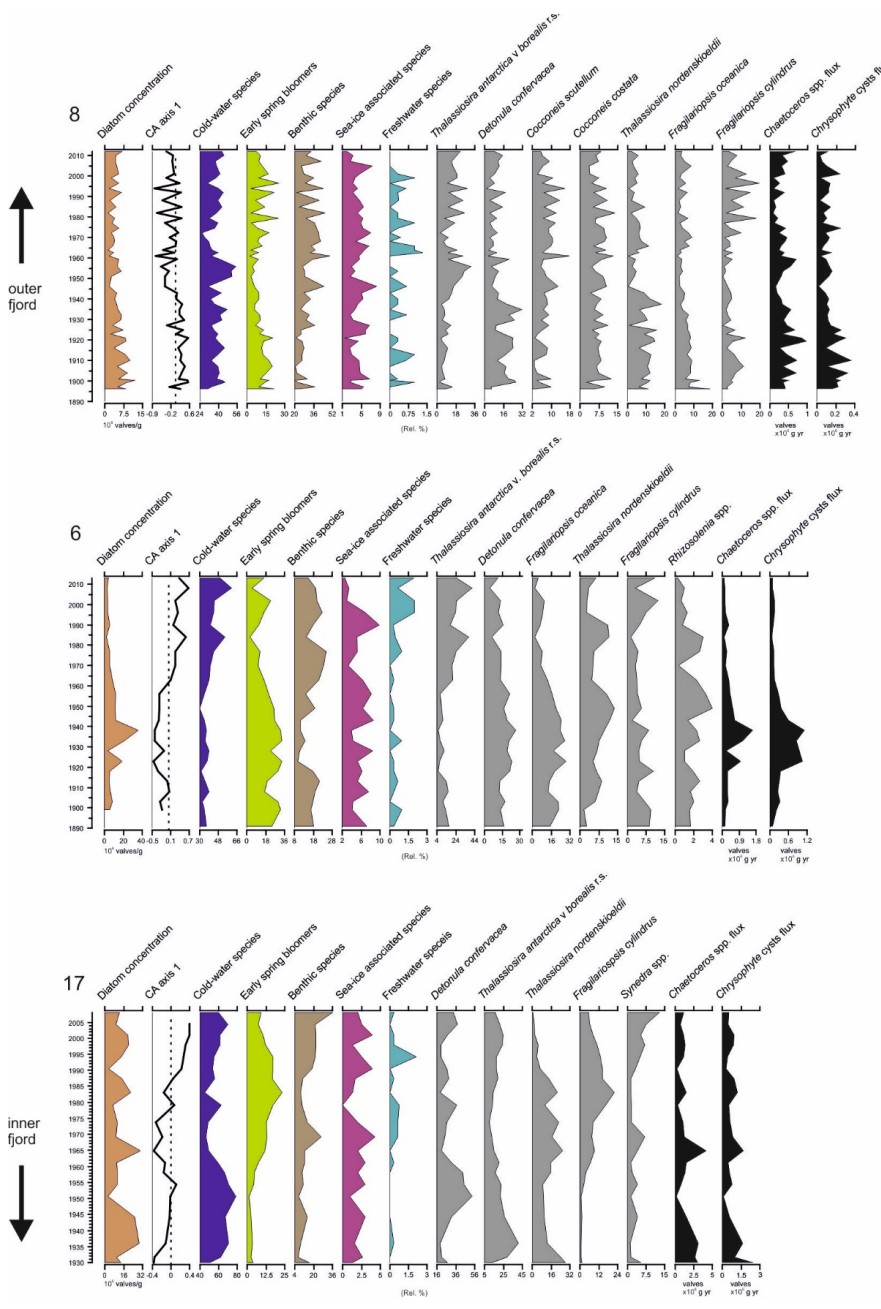

**Figure 4.** Diatom valve concentration ($10^6$ valves/g dry sediment), CA axis 1 scores (dashed lines indicates "0"), relative abundance (%) of the cold-water species, early spring bloomers, benthic species, sea-ice associated species, freshwater species, relative percentage of the most abundant species in each core, *Chaetoceros* spp., and *Chrysophyte* cyst fluxes (x$10^6$ valves cm$^{-1}$yr$^{-1}$) in sediment cores SA12-ST8-1 (8), SA13-ST6-36R (6), and DANA08-17 (17). DANA08-20 (20) was devoid of diatom valves.



**3.5 Freshwater runoff**

Total freshwater runoff estimates extracted from MAR were further divided into estimates representing inner fjord runoff,

outer fjord runoff and southern fjord runoff (Fig. 5). Runoff estimates for the southern part of the fjord system (see Fig. 1 for division of the fjord regions) are excluded since they represent part of the fjord that is not investigated in this study. The estimate for the total freshwater runoff varies between 13.0 and 49.0 Gt year$^{-1}$ (mean 23.1 Gt/year$^{-1}$), for the inner fjord runoff between 9.1 and 41.1 Gt year$^{-1}$ (mean 18.0 Gt year$^{-1}$), and between 0.7 and 2.2 Gt year$^{-1}$ (mean 1.3 Gt year$^{-1}$) in the outer fjord (Fig 5). Glacial runoff makes the largest proportion of the total runoff in Nuup Kangerlua, whereas the

runoff from tundra and rainfall into the fjord have a smaller contribution (Fig. 5).

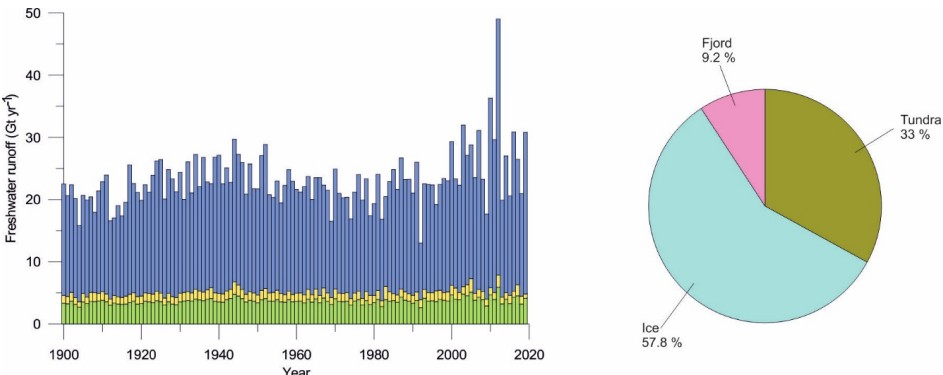

**Figure 5.** (a) Inner fjord (blue), outer fjord (yellow), and southern fjord (green) freshwater runoff estimates (Gt/yr) for 1900 – 2019. For definition of the different regions of the fjord, see red lines on Fig. 1. (b) Pie chart over the distribution of the average runoff from the ice (blue), tundra (green) and fjord rainfall (pink) between 2010 – 2019.


Correlation between the different runoff estimates and climate records were tested, showing a positive correlation between the total and inner fjord runoff and the historical air temperature record ($r = 0.61$ and $0.63$, respectively) as well as with the Fyllas Banke SST record ($r = 0.47$ and $0.49$). In contrast, the outer fjord runoff estimates did not have a statistically significant correlation to neither air nor sea-surface temperatures. All statistically significant correlations between the

studied proxies and the total, inner and outer fjord freshwater runoff, and air and sea surface temperatures are presented in Table 3.

**Table 3.** Statistically significant correlations between studied proxies (diatoms, geochemistry, grain size) and environmental variables (total, inner, and outer freshwater runoff, historical air temperatures and sea-surface temperatures `SSTs`) in sediment cores SA12-

ST8-1 (8), SA13-ST6-36R (6), DANA08-17 (17) and DANA08-20 (20). Correlation coefficient (r) values are presented in brackets for each correlation. Positive correlations are in bold.

| Site | Total runoff | Inner runoff | Outer runoff | Air temperature | SST |
|---|---|---|---|---|---|
| 8 | *Chrysophyte* cysts (-0.57) <br> *Cocconeis costata* (-0.44) <br> **Clay; <2 μm (0.42)** <br> Silt; 2-63 μm (-0.43) <br> Fine sand; 63-150 μm (-0.44) <br> **Fine grains; <63 μm (0.42)** | Freshwater species (-0.37) <br> *Cocconeis costata* (-0.41) <br> *Chrysophyte* cysts (-0.56) <br> **Clay; <2 μm (0.40)** <br> Silt; 2-63 μm (-0.41) <br> Fine sand; 63-150 μm (-0.39) <br> **Fine grains; <63 μm (0.40)** | Diatom abundance (-0.44) <br> *Fragilariopsis oceanica* (-0.39) <br> **Clay; < 2 μm (0.38)** <br> Silt; 2-62 μm (-0.43) <br> Fine sand; 63-150 μm (-0.51) <br> **Fine grains; <63 μm (0.44)** | Early spring bloomers (-0.50) <br> *Fragilariopsis cylindrus* (-0.52) <br> *Chrysophyte* cysts (-0.50) | *Chrysophyte* cysts (-0.39) |
| 6 | **BSi (0.57)** | **BSi (0.54)** | **TOC (0.52)** | **TOC (0.60)** | |





| | | | | |
|---|---|---|---|---|
| **BSi flux (0.61)**<br>**TOC (0.80)**<br>**TN (0.64)**<br>**C/N ratio (0.44)**<br>*Chaetoceros* spp. (-0.47)<br>IRD; >150 µm (-0.49) | **BSi flux (0.61)**<br>**TOC (0.80)**<br>**TN (0.63)**<br>**C/N ratio (0.42)**<br>Benthic species (-0.38)<br>*Chaetoceros* spp. (-0.48)<br>IRD; >150 µm (-0.46) | **TN (0.56)**<br>**C/N ratio (0.40)**<br>IRD; >150 µm (-0.49) | **TN (0.45)**<br>Benthic species (-0.39)<br>Chaetoceros spp. (-0.42)<br>IRD; >150 µm (-0.43) | |
| **17**<br><br>**BSi (0.67)**<br>**BSi flux (0.50)**<br>**TS (0.50)**<br>Early spring bloomers (-0.56)<br>**Sea ice associated (0.47)**<br>**Cold-water species (0.63)**<br>Freshwater species (-0.54)<br>*Chrysophyte* cysts (-0.59)<br>***Th. antarctica v borealis (0.64)***<br>*Thalassiosira nordenskioeldii* (-0.49)<br>*Fragilariopsis cylindrus* (-0.52) | **BSi (0.64)**<br>**BSi flux (0.48)**<br>Early spring bloomers (-0.58)<br>**Sea ice associated (0.46)**<br>**Cold-water species (0.64)**<br>Freshwater species (-0.57)<br>*Chrysophyte* cysts (-0.57)<br>*Th. antarctica* v borealis (0.65)<br>*Thalassiosira nordenskioeldii* (-0.46)<br>*Fragilariopsis cylindrus* (-0.53) | **BSi (0.48)**<br>**Diatom abundance (0.56)**<br>**Cold-water species (0.50)**<br>Chrysophyte cysts (-0.53)<br>***Th. antarctica v borealis (0.59)*** | **BSi flux (0.30)**<br>Early spring group (-0.73)<br>**Cold-water species (0.60)**<br>Freshwater species (-0.55)<br>***Th. antarctica v borealis (0.50)***<br>*Fragilariopsis cylindrus* (-0.68) | Early spring bloomers (-0.50)<br><br>**Cold-water species (0.54)**<br>Freshwater species (-0.45) |
| **20**<br><br>**BSi (0.76)**<br>**BSi flux (0.84)**<br>**Sedimentation rates (0.73)**<br>Fine grains; <63 µm (-0.50)<br>Clay; <2 µm (-0.69)<br>**IRD; >63 µm (0.82)** | **BSi (0.73)**<br>**BSi flux (0.81)**<br>**Sedimentation rates (0.71)**<br>Clay; <2 µm (-0.68)<br>IRD; >63 µm (0.80) | **BSi (0.66)**<br>**BSi flux (0.71)**<br>**Sedimentation rates (0.62)**<br>**Fine sand; 63-150 µm (0.62)**<br>Fine grains; <63 µm (-0.70)<br>**IRD; >63 µm (0.63)** | Clay; <2 µm (-0.50)<br>**IRD; >63 µm (0.53)** | |

## 4. Discussion

In Arctic fjords, marine primary production is a crucial ecosystem function that sustains important ecosystem services
such as fisheries and indigenous livelihoods (Berthelsen, 2014; Lydersen et al., 2014; Meire et al., 2017; Laufer-Meiser
et al., 2021). Here, we present sub-decadal records of temporal and spatial changes on marine primary production and
diatom species composition in Nuup Kangerlua since the late 19[th] century and discuss these changes in relation to annual
freshwater runoff estimates, historical air and sea-surface temperatures, and glacier front dynamics.

### 4.1 Records of marine primary production in Nuup Kangerlua since the late 19[th] century

Our primary production indicators from Nuup Kangerlua display large spatial variability between the studied sites (Fig.
3). In general, TOC and BSi values point to a relatively high productivity in Nuup Kangerlua throughout the 20[th] century,
when compared to modern values from other coastal Arctic sites (e.g., Ribeiro et al., 2017; Limoges et al., 2018a; Kumar
et al., 2016, Detlef et al., 2021). Similarly, high values have been observed on the northwest Greenland shelf during the
Holocene when climate conditions were favorable for high primary production (Limoges et al., 2020; Saini et al., 2020;
Ribeiro et al., 2021). The modern fjord hydrography in Nuup Kangerlua has been shown to maintain productivity also in
the outer fjord, as biomass and nutrients are partly advected from the inner fjord by currents, wind and tidal mixing which
together with the warm, nutrient-rich waters carried by the WGC, can increase the seasonal productivity (Juul-Pedersen
et al., 2015). Also, the year-round ice-free conditions in the outer fjord may prolong the productive season, whereas glacial
calving and sea-ice cover in the inner fjord shorten the productive window.

Today, spatial trends in primary production in Nuup Kangerlua are mainly controlled by sea-surface temperature and
salinity gradients along with nutrient availability, so that the highest productivity rates have been reported from the inner
fjord area (Krawczyk et al., 2015b, 2018; Meire et al., 2016b, 2017). In contrast, our sediment records with the highest
organic carbon content were retrieved from the outer fjord (site 8), where average TOC and BSi levels are up to 46- and





3-times higher, respectively, than in the inner fjord area (Fig. 3). While sedimentation rates are more than 2-times higher
in the inner fjord, mean BSi fluxes are still slightly higher at the outer fjord site than at the two inner fjord sites 17 and 20
(Table 2). Such an inner-outer fjord gradient of increasing productivity can be found elsewhere along the Arctic coasts
(Arendt et al., 2016; Kumar et al., 2016; Ribeiro et al., 2017; Limoges et al., 2018a). Whereas productivity at the outer
fjord has remained relatively stable (and high) since the late 19[th] century, this site records frequent low-amplitude
fluctuations occurring at a sub-decadal scale.

The diatom productivity (absolute concentrations and fluxes) is highest at site 17, and this part of the Nuup Kangerlua
fjord is characterized by very high rates of annual primary production (up to 120 g C m$^2$ year$^{-1}$) according to data from
the Greenland Ecosystem Monitoring Programme (Juul-Pedersen et al., 2015; Meire et al., 2015). Diatoms alone account
for about 95 % of the biomass here (Arendt et al., 2011; Meire et al., 2016a). The high productivity in this part of the
fjord has been suggested to be maintained by the high dissolved silica (DSi) input from surface runoff combined with
upwelling of nutrient-rich deep waters (Meire et al., 2016b, 2017). The high marine productivity at site 17 can be
explained by its ideal location receiving freshwater discharge from both surface runoff and several marine-terminating
glaciers (Fig. 1). Subglacial discharge plays a key role in sustaining high productivity in this part of the fjord. Although
the most glacier-proximal site (20) has a relatively high BSi flux (9.2 mg cm$^2$ yr$^{-1}$ on avg.), no diatom valves were found
apart from a few in the uppermost centimeters of the sediment core. The near absence of diatoms in this record may be
explained by dissolution of the diatom valves in the water column or water-sediment interface due to adverse
physicochemical conditions (Ryves et al., 2006).

**4.2 Sources of sedimentary organic matter**

The stable isotope carbon-13 ($\delta^{13}$C) and the carbon to nitrogen (C/N) ratio can be used to trace whether the organic matter
in the sediment originates from marine or terrestrial sources. Autochthonous organic carbon is produced by the primary
producers using the $CO_2$ dissolved in the seawater, whereas allochthonous organic carbon sources can be of terrestrial or
marine (e.g. sea ice) origin (Bianchi et al., 2020). At sites 8 and 6, the organic matter in the sediment can be traced to be
of marine origin, as the measured $\delta^{13}$C values (between -20 ‰ and -22 ‰) are typical for protein-rich marine
phytoplankton, whereas the terrestrial organic carbon would generally be more depleted in $\delta^{13}$C. In our record, we find
no evidence of freshwater runoff increasing the amount of terrestrial carbon in the sediment, as the variations in $\delta^{13}$C
values are minimal and show no increase in pace with the freshwater runoff dataset (Fig. 3). Despite the C/N-ratio from
site 6 having a positive correlation with freshwater runoff, the very low values (average 6.3 at site 8 and average 7.4 at
site 6) further support autochthonous organic matter sources, as C/N-ratios above 20 would indicate terrestrial carbon
sources (Meyers, 1994). However, in marine sediments, bound inorganic nitrogen can constitute a large portion of the
total nitrogen concentration and thus the marine organic matter might be more depleted in $\delta^{13}$C than terrestrial organic
matter (Kumar et al. 2016). The C/N-ratios measured in this study were not corrected for the land-derived inorganic
nitrogen and thus might be underestimated. These analyses ($\delta^{13}$C and C/N-ratio) were performed only for the glacier-
distal records (8 and 6) and not for the glacier-proximal records (17 and 20) due to extremely low carbon contents in these
sediments (Fig. 3). Although the sedimentary organic carbon in our records is mainly from marine sources, we found
freshwater diatoms (Fig. 4) that originate from terrestrial freshwater sources e.g., lakes and ponds, and enter the fjord via
runoff (Jensen, 2003). Overall, the relative abundance of freshwater diatoms is extremely low (<2 %), but it represents a



clear fingerprint of terrestrial origin as these species generally have very low to zero tolerance to saline waters and are not found in marine environments (Cremer, 1998; Foged, 1973).

**4.3 Freshwater discharge positively impacts primary production in the inner fjord**

Our proxy data display a substantial increase in marine primary production at the glacier-proximal sites (17 and 20) from

ca. 1995 onwards, as sedimentary BSi contents increase rapidly in pace with the inner fjord freshwater runoff (Fig. 6). In modern monitoring studies the inner fjord is found to display high productivity, as sub-glacial freshwater discharge and the subsequent meltwater plume lead to upwelling of nutrients from the deep water into the surface layer (Meire et al., 2016b, 2017; Hopwood et al., 2018, 2020). The modern monitoring data from Nuup Kangerlua reaches back to 2008 (Greenland Ecosystem Monitoring), however, our long-term records suggest that the high productivity levels, prevailing

at the glacier-proximal area today, initiated already in the 1990´s. The strong positive correlation between marine productivity and freshwater runoff (Table 3) shows that increased productivity in this fjord is tightly associated to freshwater discharge. This positive correlation suggests that in the future, increased freshwater discharge is likely to have a positive socio-economic impact, as high productivity is reflected in the higher trophic levels such as fish and marine mammals (Boye et al., 2010). For example, halibut fisheries, which are economically important for Greenland

(Berthelsen, 2014), have been shown to have a positive correlation with meltwater input (Meire et al., 2017).

Despite most of the freshwater originating from glacial discharge (Fig. 5), the increasing relative abundance of freshwater diatoms recorded from the 1990´s (Fig. 6) indicates flushing of freshwater deposits (such as lake Tasersuaq) into the fjord. Coarse sediment grains (>63 μm) at site 20 display a strong positive correlation with freshwater runoff (Table 3) suggesting that freshwater input to this site originates from solid ice discharge. The coarse grain size in the sediment

generally represents the IRD deposited by icebergs which typically release most of the coarse grains close to the source as they melt. The melting of icebergs can cause water column vertical mixing in a similar way as subglacial discharge, and thus provide more nutrients to the surface waters (Kanna et al., 2018). Also, sedimentation rates are increasing at sites 17 and 20 along with freshwater runoff (Table 3), suggesting that a large amount of sediment is being delivered to the fjord with freshwater discharge. Sub-glacial discharge typically causes sediment suspension plumes (Fig. 1) which

add nutrients from deep waters to the surface for the primary producers (Meire et al., 2017; Hopwood et al., 2018, 2020). An increase in freshwater discharge accompanied by increased sediment fluxes and upwelling, would represent a shift from clear, stratified waters to turbid, mixed waters after the 1990´s. In fact, our diatom record shows that species such as *Thalassiosira nordenskioeldii* and *Fragilariopsis cylindrus*, as well as *Chrysophyte* cysts, that prefer low-salinities and stratified waters (Hasle and Heimdal, 1998; von Quillfeldt, 2004; Krawczyk et al., 2015a, 2015b), decrease after the

1990´s (Fig. 4), while the number of benthic species, associated to water turbidity and to low light conditions (Glud et al., 2002; Karsten et al., 2011) increases (Fig. 4).

In contrast to the inner fjord, the record from the outer fjord site (8) shows no correlation between productivity and freshwater runoff throughout the entire studied period, and here, freshwater runoff is limited (Fig. 5). The diatom CA axis suggest an important change in the assemblage around the 1940´s (Fig. 6), from an assemblage dominated by early-spring

bloomers to an assemblage dominated by benthic species and late-summer bloomers, e.g., *Thalassiosira antarctica* v. *borealis* r.s. (Fig. 4). This suggests an increase in water turbidity, and/or more nutrient-limited conditions (Glud et al., 2002; Karsten et al., 2011). However, the timing of this shift in the assemblage composition is not associated with any

significant changes in the freshwater runoff and may therefore be caused by changes in e.g., wind strength and consequently mixing of the water column.

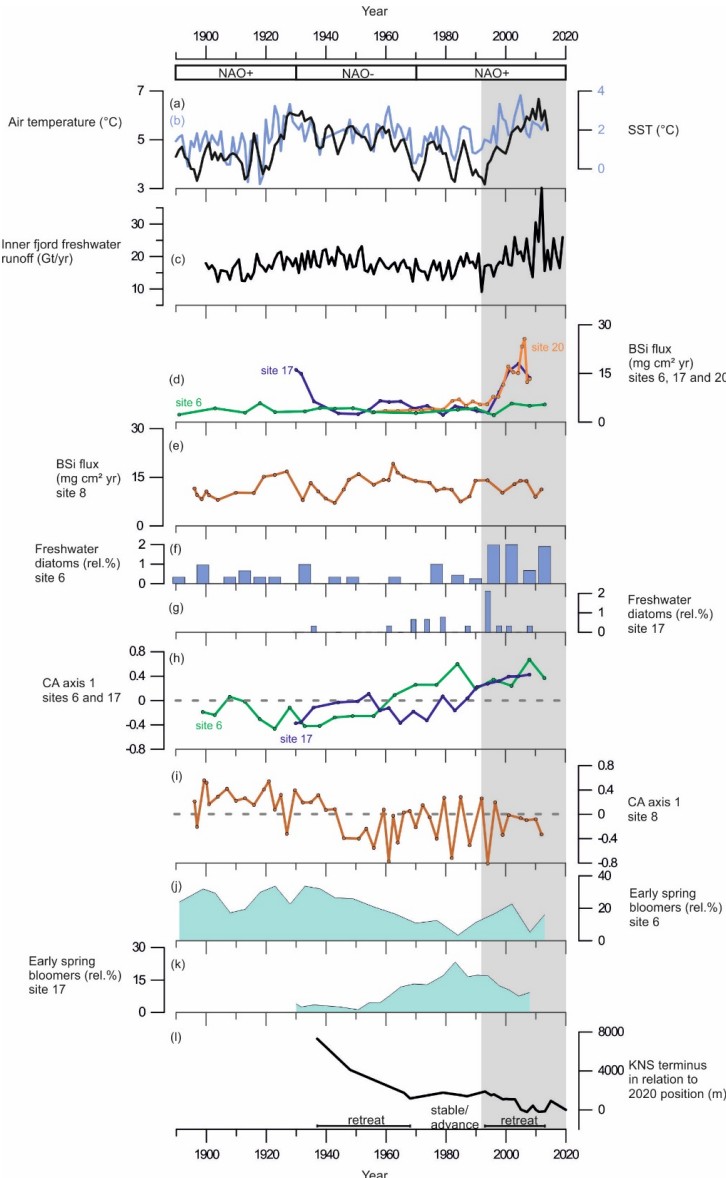


**Figure 6.** (a) 3-year average of annual summer (May, June, July, and August) air temperature measured at Nuuk airport (black line) from Ribergaard, 2014 and (b) June sea surface temperatures; SSTs (0 – 40 m) from Fylla Bank (blue line) from Ribergaard, 2014. (c) Inner fjord freshwater runoff estimates (Gt/yr). (d) BSi flux (mg cm$^2$ yr$^{-1}$) at site 6 (green line), site 17 (purple line) and site 20 (orange line). (e) BSi flux (mg cm$^2$ yr$^{-1}$) at site 8. (f) Relative percentage of freshwater diatoms at site 6. (g) Relative percentages of freshwater

diatoms at site 17. (h) Diatom CA axis 1 scores at site 6 (green line) and site 17 (purple line), dashed lines indicate "0". (i) Diatom CA



axis1 scores at site 8, dashed line indicates "0". (j) Relative percentage of early spring bloom species at site 6. (k) Relative percentage of early spring bloom species at site 17. (l) Changes in KNS glacier front position (m) in a relation to observed position in 2020. Shaded area marks the period from 1995 to present.

### 4.4 Freshwater discharge influences spring and summer bloom dynamics

Based on modern observations, the annual marine primary production in the Nuup Kangerlua fjord is sustained by two seasonal blooms of similar magnitude occurring during the early spring and the late summer (Juul-Pedersen et al., 2015). Our long-term perspective indicates that the dynamics and magnitude of these blooms have varied according to freshwater discharge (Fig. 6). We find that freshwater runoff leads to a stronger spring bloom signal at site 6, while promoting late-summer bloomers at site 17 (Fig. 4 and 6), and vice versa during periods of lower runoff. In terms of species responses,

the diatom records show that species blooming in early spring during the ice melt (*Fragilariopsis* spp.) or right after ice break-up in late spring (*Thalassiosira nordenskioeldii*) (Cremer, 1998; Jensen, 2003; Oksman et al., 2019) dominate the assemblage at site 6 during periods of higher freshwater discharge. Yet, the species blooming during the late summer (*Detonula confervacea* and *Thalassiosira antarctica* var. *borealis* r.s.) as well as *Chaetoceros* spp., which typically follow nutrient exhaustion and represent the last stage of the productive season (Krawczyk et al., 2015a), are more common at

site 17 during periods of increased freshwater runoff (Fig. 4).

Modern observations have shown that today spring and summer bloom dynamics are mainly controlled by seasonal changes in salinity and temperature as well as fjord hydrography, and tidal and wind mixing (Juul-Pedersen et al., 2015; Krawczyk et al., 2015a, 2018; Meire et al., 2016a). Other environmental factors, in particular sea ice, might better explain the differences observed at the two sites, as sea ice can also influence the timing and magnitude of the spring bloom

(Meire et al., 2016a). At the glacier proximal site (17), a relatively high number of sea ice species (which have a positive correlation with freshwater runoff) and low number of early spring bloomers suggest sea-ice presence continuing late into the spring limiting the spring production season (Fig. 4). However, the substantially high percentage (up to 79 % in the 1950´s) of late summer bloomers alongside with elevated BSi and diatom flux (Fig. 3) are indicative of a productive summer season.

Spring bloom is a significant part of the annual primary production in Nuup Kangerlua, (Juul-Pedersen et al., 2015) and increased spring production in the Arctic is a typical feature of the ongoing climatic warming, which has been evident in the Arctic Ocean for the recent decades (Renaut et al., 2018). The length of the productive season has implications for $CO^2$ capture as the fjords can function as carbon sinks (Meire et al., 2015; Rysgaard et al., 2012). In the future, increasing freshwater discharge is likely to extend the productive season despite the timing of the blooms occurring in different parts

of the fjord system. The spring bloom is likely to occur earlier in the side branch of the fjord, whereas the magnitude of the late summer bloom increases along the main ice flow pathway.

### 4.5 Freshwater impact reaches into the glacial-distal side branch

Until now, the impacts of freshwater runoff on primary production have mainly been studied in the main branch of the Nuup Kangerlua fjord system (Juul-Pedersen et al., 2015; Meire et al., 2016a, 2016b, 2017; Krawczyk et al., 2018),

whereas productivity changes in the side branches are less known. Today, the upwelling of nutrients by subglacial processes in Nuup Kangerlua is suggested to affect primary production downstream, to a distance of 10 – 100 km along the pathway of the advected bloom (Meire et al., 2016a; Hopwood et al., 2018). Here, we show that freshwater runoff



impacts primary production in the glacier-distal site at the side branch of the fjord system (site 6). Here, both BSi flux and TOC strongly correlate with the freshwater runoff, despite this site being in the sheltered side branch ca. 110 km from the Kangiata Nunaata Sermia (KNS) termini and not directly on the main flow path of glacier influence (Fig. 1). The high number of early spring bloomers at site 6 indicates an earlier seasonal ice breakup and strong inflow of freshwater into the side branch as these species thrive in cold and fresh surface waters. Diatom production is high (up to 36 x $10^6$ valves / g) and a flux of 7.2 valves cm$^{-1}$yr$^{-1}$ during the 1930´s and the 1940´s while freshwater runoff was elevated, and sea-surface temperatures are relatively high (Figs. 3 and 6). The high productivity is further supported in the diatom species composition data by the *Rhisozolenia* genera as it reaches the highest abundances of the record around 1945 (Fig. 4). This genus has been linked to a nutrient-rich and stratified water column, conditions that are generally related to meltwater pulses (Kemp et al., 2006).

**4.6 Long-term variability in productivity links to atmospheric forcing**

Teleconnections in the North Atlantic region, such as the North Atlantic Oscillation (NAO) and the Atlantic Multidecadal Oscillation (AMO), have been shown to drive marine primary production (Martinez et al., 2009). These decadal-scale ocean-atmosphere oscillations modulate sea-surface conditions through fluctuations in temperature, precipitation, and wind, and thus, impact ecosystem responses (Martinez et al., 2009; Racault et al., 2017). In west Greenland, the locations of glacier termini are driven primarily by atmospheric temperatures reflecting NAO phases, and atmospheric forcing has been defined as the primary driver of KNS ice margin changes (Lea et al., 2014; Bjørk et al., 2018). NAO phases modulate the climate so that during a negative NAO phase West Greenland experiences warmer climate, and increased precipitation, whereas during the positive phase the opposite occurs (e.g., Hurrell, 1995; Bjørk et al., 2018).

In Nuup Kangerlua, most of the glacier termini responded to the warm atmospheric temperatures during the negative NAO phase between the 1930´s and 1970´s by retreat and mass loss (Fig. 7), with KNS having the largest retreat of ca. 8 km (Fig. 6). Our record shows increased marine productivity at site 6 during this negative NAO phase (Fig. 6), and here the TOC has a strong positive correlation with regional air temperatures and freshwater runoff. Diatom record from site 6 show freshening of surface waters as *Chaetoceros* spp. and *Chrysophyte* cyst fluxes are highest together with higher relative percentage of early-spring bloomers (Fig. 4). Together, this suggests that fjord productivity responds to decadal ocean-atmosphere oscillations. After the 1970´s, the NAO phase has remained mainly positive but interestingly since the 1990´s the accelerated climate warming has over-ruled the NAO modulated climate signal (Seidenkrantz et al., 2009), and both temperatures and freshwater discharge have increased substantially despite the ongoing positive NAO phase (Fig. 6). During the recent decades, asynchronous behavior of the KNS and NS has been attributed to subglacial melt (Motyka et al., 2017), however, both KNS and NS have retreated (Fig. 7) resulting in increased freshwater discharge and productivity in the inner fjord sites. The accelerated warming of the recent decade may lead to substantial retreat of KNS in the future (Aschwanden et al., 2019). This can have a negative impact on productivity if the grounding depth changes significantly (reducing the fertilizing effect of nutrient upwelling) or if the ice margin retreats on to land. The retreat of marine-terminating glaciers is likely to have a strong impact on productivity, due to the key role that subglacial discharge plays in sustaining high fjord productivity both on interannual (Meire et al., 2017; Hopwood et al., 2018) and multi-decadal timescales as shown here.

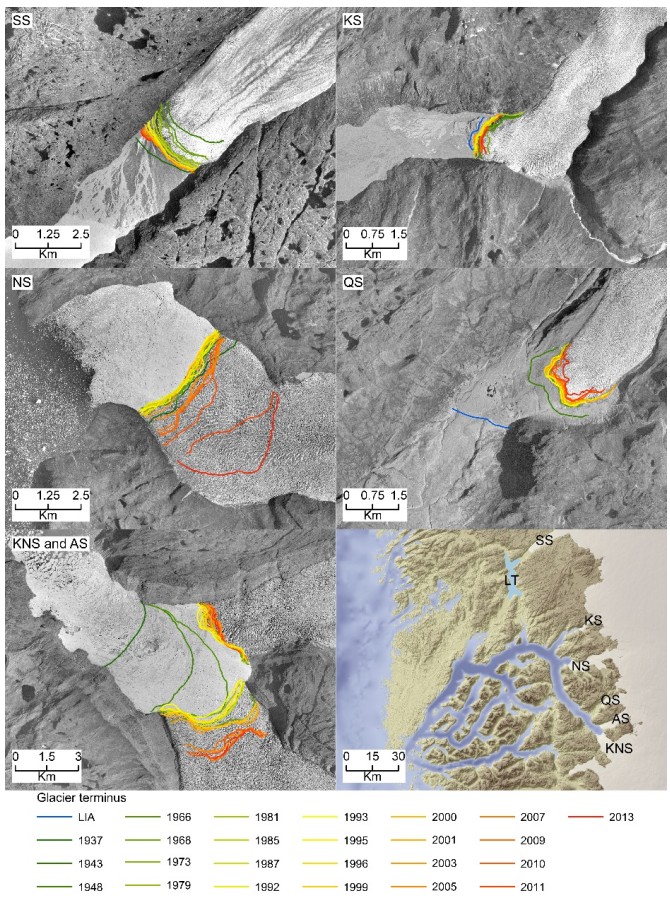

**Figure 7.** Glacier terminus changes of Saqqap Sermia (SS), Kangilinnguata Sermia (KS), Narssap Sermia (NS), Qamanaarsuup Sermia (QS), Kangiata Nunaata Sermia (KNS), Akullersuup Sermia (AS), and Lake Tasersuaq (LT) in a relation to their 2020 position.

## 5. Conclusions

Long-term records of marine productivity are needed to define the natural baseline against which to assess recent cryosphere and coastal ecosystem changes. More accurate baselines ultimately provide more robust projections of future changes. Here, we have combined sub-decadal records of primary productivity changes with estimates of freshwater runoff since the late 19th century to determine the impacts of freshwater on fjord productivity for Nuup Kangerlua, southern West Greenland.

We find that in the glacier-proximal area BSi fluxes are tightly associated with freshwater runoff, and that freshwater imposes positive impacts on marine productivity. Our record shows non-uniform responses to freshwater inside the fjord system which highlights the importance of marine-terminating glaciers to marine productivity. However, we find that the impacts of freshwater discharge are wider than previously known, as freshwater has a positive impact on the marine productivity in the glacier-distal side branches of the fjord system outside of the main discharge path flow. We conclude



that productivity is higher today than at any other time since the late 19th century. A substantial increase in productivity initiated in the 1990´s in the inner fjord due to the acceleration in the climate warming and the following cryosphere

changes (retreat in the glacier front line and sub-glacial freshwater input).

Our diatom assemblage data shows that freshwater modulates the location and magnitude of the spring and late summer blooms, and that these blooms are not synchronous inside the fjord. Freshening of the inner fjord prolongs the late spring sea-ice conditions leading to more extensive spring bloom in the side branch while summer blooms becomes larger in the inner fjord.

In the future, freshwater runoff is expected to increase and will likely result in higher marine productivity with positive influence on Greenlandic fisheries and other harvestable marine resources. It is likely that, in a warmer future with more ice-sheet melt, the fjord system may become a greater $CO_2$ sink, as increasing freshwater runoff is related to increasing marine production and a prolonged annual productive season due to increased nutrient availability. Ultimately, the future productivity of this system will be tightly linked to the fate of its marine-terminating glaciers.

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

**Author contribution**

M.O. planned the study together with S.R. S.R. designed the project (*Greenshift*) and provided sediment core material
together with N.N.P., M.-S.S., and N.M. M.O. conducted diatom, grain size, and BSi analysis, and A.B.K. assisted with BSi and grain-size analyses. T.J.A. carried out [210]Pb measurements. S.H.L. and W.T.C. provided freshwater runoff calculations for 1900-2010 together with K.D.M. who provided freshwater runoff data. K.K.K. provided data for ice margin position. M.O. wrote the manuscript with input from all the co-authors.

**Acknowledgement**

This study received financial support from Geocenter Danmark (project *Greenshift* grant 2018-5 to Sofia Ribeiro). The 2013 cruise with RV *Sanna* was funded by the Arctic Research Centre of Aarhus University, Denmark. SRI received additional funding from the Independent Research Fund Denmark (grant no. 9064-00039B). MSS was funded by the Danish Council for Independent Research (grant no. 7014-00113B (G-Ice) and 0135-00165B (GreenShelf) with
additional funding from the European Union´s Horizon research and innovation program under Grant Agreement no. 869383 (ECOTIP). We thank Ida Sofie Mikaelsdóttir Olsson, Annette Ryge and Charlotte Olsen for help with biogenic silica analysis and diatom slide preparation. Finally, we thank Thomas Juul-Pedersen for accommodating and supporting our sediment coring activities during the Greenland Ecosystem Monitoring cruise with RV *Sanna* in 2012.