# Peer review of "Impact of freshwater runoff from the southwest Greenland Ice Sheet on fjord productivity since the late 19th century"

_The Cryosphere, 2021_

## Author Comment (AC1)

Response to Referee#1 comments:

We thank Referee#1 for reviewing our manuscript, and we appreciate the positive and constructive comments provided which will help us to improve our manuscript. We have addressed all the comments and our detailed response is below (in blue) after each comment.

It was my pleasure to review the preprint titled *"Impact of freshwater runoff from the southwest Greenland Ice Sheet on fjord productivity since the late 19th century"*. Oksman et al. present an interesting assessment of the impacts of freshwater on fjord productivity through the combination of modelled freshwater runoff estimates and a multi-proxy approach to constrain primary productivity. The authors focus on Nuup Kangerlua, south-west Greenland, which has been well studied, particularly in terms of productivity, but the approach of combining sediment records for the 19th century to present with modelled runoff data is novel and provides an alternative perspective into the important questions of fjord biogeochemical cycling. The manuscript is well-written, with a logical and thorough discussion and I would recommend it for publication after minor revisions, if the authors can address my main concern relating to their interpretation of BSi concentrations (detailed below in "General Comment 1"). I have detailed two main comments on the discussion below, followed by line-by-line specific comments and technical, editorial comments.

General Comment 1:

The definition of primary productivity could be made clearer in several parts of the manuscript, to ensure the readers can easily identify how the authors are assessing the patterns of primary productivity change. The authors do outline the multiple proxies used to reconstruct primary productivity in Line 133, but it could be useful to highlight this again within the discussion so it is clear to the reader which proxies are being used to define primary productivity. For example, in Section 4.3 should the reader assume all of the proxies mentioned in line 133 (i.e. diatom fluxes and assemblage composition, BSi content, composition and origin of organic matter and grainsize analysis), or just specific proxies are being used during the assessment of correlations between productivity and freshwater discharge. Table 3 shows that BSi is the only productivity proxy that has a significant correlation with total runoff in sites 6, 17 and 20, suggesting that this is the proxy that the authors are placing the most emphasis on to form the discussion in section 4.3.

We have now added a definition of primary productivity in the introduction: "the process by which primary producers convert solar energy into organic compounds" and specified in the text which proxies we use to infer past changes in productivity over time. The discussion section 4.1. now begins with "Our productivity indicators (diatom fluxes, BSi, and TOC) from Nuup Kangerlua…".

This requires some explanation from the authors into their methodology for BSi concentration analysis. The authors used the DeMaster (1991) extraction method, which can overestimate BSi content due to the simultaneous dissolution of non-biogenic Si fractions (e.g. Barão et al. 2015, Pickering et al. 2020). This is particularly problematic in fjord environments where glacial meltwaters can deliver substantial quantities of glacial flour in freshwater fluxes, containing potentially high concentrations of subglacially-derived non-biogenic, amorphous silica (ASi), which would also dissolve during the $Na_2CO_3$ extraction employed by the authors (e.g. Hawkings et al. 2017, Hatton et al. 2019). Therefore, the reported elevated BSi concentrations are likely also a reflection of increasing ASi concentrations when glacial meltwater input increases. This makes the reliance on the concentration data potentially problematic for inferring changes in primary productivity linked to freshwater discharge, as it appears that the authors are unable to deconvolve BSi and ASi concentrations in their sediment records. The authors note there is a large amount of sediment delivered to the fjord with freshwater discharge (line 414), and it is this sediment that I would expect to contain substantial ASi concentrations, which would impact the "BSi" concentrations that the authors report currently. While the multi-proxy approach of the study helps with this problem in part and I completely expect high BSi

concentrations based on previous studies of productivity within this fjord, I think it could be important for the authors to address the potential of ASi within their reported BSi concentrations, especially considering the BSi concentrations appear to form the largest portion of evidence for increasing or decreasing productivity within the current discussions.

We acknowledge that this is a valid concern as marine-terminating glaciers are a large contributor to Si fluxes in fjord systems. When calculating BSi concentrations in the samples, we corrected our measurements for mineral dissolution, and we have now clarified this further in the methods section. We would also like to emphasize that we use a wet-alkaline extraction method on a weak base solution (1% $Na_2CO_3$) which minimizes the extraction of Si from clays (Conley, 1998) compared to a strong base extraction (NAOH) that is more prone to dissolve clay minerals (Barao et al., 2015). The overestimation of BSi because of non-biogenic Si is a risk especially when using a strong base method on samples with low BSi concentrations (DeMaster, 1991).
How much non-biogenic Si contributes to BSi estimates depends also on the composition of the sample and reactivity of Si (Barao et al., 2015), and this has been shown to be significantly problematic in sediments with pedogenic Si. We want to stress that this is not relevant for the samples from glacial environments as in our study.
In addition, fragments of diatom valves were found in the upper part of the core from site 20 (as discussed in section 4.1). This shows that this site, while presenting the lowest concentrations of BSi in this study, has not been totally barren from life and diatoms were present yet complete valves did not preserve through the water column into the sediments. Having remains of diatoms at the top of the sediment core where BSi content increases strengthen our statement that measured BSi represents Si produced by biological processes.
In summary, given that our samples were corrected for mineral dissolution and we selected the wet-alkaline extraction method on a weak base, we are convinced that minerogenic Si is not a significant contributor in our measurements.

Barão, L., Vandevenne, F., Clymans, W., Frings, P., Ragueneau, O., Meire, P., Conley, D.J., and Struyf, E., 2015. Alkaline-extractable silicon from land to ocean: A challenge for biogenic silicon determination, Limnol. Oceanogr. Meth., 13(7), 329-344.
Conley, D.J., 1998. An interlaboratory comparison for the measurement of biogenic silica in sediments. Mar. Chem. 63, 39-48.
DeMaster, D.J., 1991. Measuring biogenic silica in marine sediments and suspended matter, in: Marine Particles: Analysis and Characterization, Geophysical Monograph 63, American Geophysical Union, edited by: Hurd, D.C. and Spencer, D.W., Washington, D.C., 363-367.

General Comment 2:

I was a little disappointed with the lack of discussion into how and why Site 20 was found to be barren of diatoms. This appears to be a very surprising and significant finding that warrants more than 1 sentence of basic speculation (line 369). I would ask the authors why the conditions at this site would be significantly different when compared to Site 17 (approximately 25km down-fjord) to lead to diatom valve dissolution? Could the lack of diatoms in the record reflect a true finding and highlight low primary productivity at this site, instead? If the reported BSi concentrations actually reflect glacially-derived ASi, then the correlation between BSi and freshwater input at this site could just reflect a delivery of glacial flour from subglacial discharge, rather than a link to primary productivity. I think it could be useful for the authors to consider the lack of diatoms in the record at this site a little more within the discussion to ensure the different potential fjord processes are reflected within the discussion.

We have added a more detailed discussion of site 20 diatoms: "In general, biogenic silica contents at site 20 are significantly lower until the late 1990s, suggesting low primary production here, compared to other sites. The major increase in BSi and appearance of diatom valve fragments in the upper part of the core indicate an increase in productivity, although remaining significantly lower than at the other sites. The absence of diatoms

and low BSi values indicate, in turn, very low productivity levels. The near absence and overall poor preservation of diatom valves may also be explained by dissolution of the diatom valves in the water column or water-sediment interface due to adverse physicochemical conditions (Ryves et al., 2006)".

We would also like to comment that glacial flour at this site would most likely be reflected in the sediment record as a significant increase in clay particles. However, the BSi content (and flux) and coarse sediment grain size (>63µm) at site 20 have a strong correlation with freshwater discharge, while fine grain sizes (<63µm) and the sediment clay content have a negative correlation with freshwater discharge. This suggests that the BSi increase here is more related to ice calving processes rather than a sub-glacial plume event, which is more evident in the site 17 record.

**Specific Comments**

Line 90 (Fig. 1): Is the true colour imagery from the time of sampling or just the most recent image from Sentinel-2? It may be useful to show the satellite image from a timepoint close to when samples were collected, as sea ice cover in the fjord is likely to vary seasonally?

All cores were taken during the summertime, and thus we selected Sentinel-2 satellite image from August 9th, 2018 to represent summer season conditions. The purpose of the satellite image is to show the difference between the studied sites (glacial drift, water turbidity) rather than represent the conditions during the time of sediment coring.

Line 153: It could be useful to label Fig.1 with the site numbers alongside the core identifiers for easier reference.

We have added site numbers in Figure 1.

Line 260 (Table 1): Could the units be placed in the column headings instead of in each cell?

We have moved units into the column head.

Fig. 3 and Fig. 4: It could be misleading to the reader to label the plots Outer à Inner fjord, as two sites are within a fjord branch and not part of the main fjord system, so it's not a simple transect being visualised in these plots.

We have removed arrows pointing to outer/inner fjord, and instead use arrows to point glacier proximal/distal.

Line 330 (Table 3): Do the authors have any hypotheses into why freshwater species negatively correlate to total runoff and inner runoff at site 17? This could be an interesting finding to discuss.

This is very interesting observation, as we would assume these species to be strongly related to freshwater runoff as they do not habit marine environment but are transported to marine environment from freshwater sources. Freshwater species are present in site 17 record throughout the core but peak in the mid 1990´s. The presence of freshwater species pre-1990´s is low and could be linked to lake outburst events depositing diatoms near the site rather than overall freshwater trends. We want to avoid concluding too much on the freshwater species, as they represent less than 3% of the assemblage in site 17.

Line 340: It would be useful to define the "primary production indicators" here again for the reader, so that it is clear throughout this section how "high productivity" is judged. Also, see General Comment 1, regarding the use of BSi in the assessment of productivity in the fjord environment.

We have clarified throughout the text which productivity proxy we discuss.

Line 341: Some exemplary values from this study and the referenced published data could be useful here for a quick comparison, rather than the reader having to refer to the original publications.

We have added values from the reference data.

Line 370: See General Comment 2. Some further explanation of this result would be useful here. The previous discussion focuses on how the inner fjord environment has the highest productivity, yet the inner-most sampling site is barren of diatoms, which contradicts this assertation and many previous studies. I struggle to understand why the local conditions at this site would lead to the dissolution of diatom valves, whereas they were well preserved at site 17. While I agree with the authors and would suspect there is high productivity in this area of the fjord, at least for part of the melt season, I think it is important that the authors address the surprising result of no diatoms within the sediment record in more detail.

Previous studies from this fjord have focus on modern time using water samples or surface sediment samples. From the diatom assemblage living in the surface waters, only ca. 10% make it into sediment record, and thus it can happen that diatoms are missing from the sediment records in sites which are shown to have diatoms in the water column. We found diatom valve fragments in the upper part of this which confirms that diatoms have habitat this site, yet diatoms were not abundant enough to make meaningful assemblage counting. We have added more discussion (see our reply to General comment 2) about the (missing) diatom record and productivity in site 20. We have also clarified in the text that it is the site 17 which holds high productivity in the inner fjord area.

Line 372: Should this heading instead read, Sources of sedimentary organic matter of distal sites"? Without $\delta^{13}C$ and C/N analysis of the two proximal sites, this section can only describe the source of organic matter at two sites. Would the authors hypothesise that there would be a higher proportion of terrestrial organic carbon, and thus more depleted $\delta^{13}C$ values, if analysis was possible?

We have changed the heading to "Sources of sedimentary organic matter in glacier-distal sites" and added hypothetical speculation that organic matter in the inner fjord could have terrestrial origin (see reply for comment on line 389).

Line 377: Specify here that this is only for sites 6 and 8, as it currently appears that this is true for all sites, but you do not have data to assess the source of organic matter at the two proximal sites.

We have specified this in the text.

Line 389: Is this just for sites 6 and 8? Was the proportion of freshwater species higher at site 17, which would indicate the impact of terrestrial sources is greater at this site, which could be expected. This data could potentially allow the authors to make a statement about the organic matter source of the proximal sites, with the clear caveat that $\delta^{13}C$ and C/N analysis could not be completed.

We have clarified that freshwater species were found in sites 8, 6, and 17. And based on presence of freshwater at site 17 it is not justified to hypothesize that organic carbon could be from terrestrial origin as site 6 has higher % of freshwater species, and there organic carbon is from marine sources.

Line 400: Which indicators are you using as "marine productivity" proxies in this statement, please be list them here so the reader can be clear.

We have specified the marine productivity proxy discussed here, as well as in other parts of the manuscript.

Line 422: This sentence opening could be misunderstood, as it begins by contrasting the outer and inner fjord. However, Site 6, which the authors also label as an 'outer site' does have correlations between runoff and BSi and TOC. Could the lack of correlation at Site 8 be a result of the sample location being within a fjord side-branch, rather than the main fjord where the freshwater discharge flows? Naming this site as the outer fjord suggests that it is linked to the main branch of the inner fjord, which is unlikely.

We have modified text here and removed comparison between inner versus outer fjord and discuss site 8 as glacial-distal site rather than outer fjord site.

Line 444: It is not clear from Fig. 6 to link increased freshwater to differences in bloom timing between sites 6 and 17, as directly comparing the plots is difficult. Could the authors highlight these patterns more clearly? By eye, it even appears that there is decline in early spring bloomers at Site 6 coinciding with the large peak in freshwater around 2012, which is the opposite pattern to that described. The authors could also reference the negative correlation between early spring bloomers and freshwater runoff at Site 17 from Table 3 here.

We have made this comparison clearer and added a reference to Table 3.

Line 449: Including a plot of freshwater runoff in Fig. 4 could be helpful to allow for easier comparison between the pattern of changes in freshwater runoff and relative species types.

We have added a plot of freshwater discharge in Figure 4.

Line 533: Could include a statement that the likelihood of fjords becoming a greater $CO_2$ sink will only occur while glaciers are discharging directly into the fjord and have not retreated beyond the grounding line.

We have added this.

**Technical Corrections**

Line 20: Avoid the use of "present" twice in one sentence. Corrected

Line 269 (Fig. 3): Axis labels are too small to be legible at 100% view, please edit the figure so it can be viewed more easily. Figure and fonts have been enlarged.

Line 304 (Fig. 4): Axis labels are too small to be legible at 100% view, please edit the figure so it can be viewed more easily. Figure and fonts have been enlarged.

Line 316 (Fig. 5): Change the label of the pink section of the pie chart from 'fjord' to 'rainfall'. Changed.

Line 463: Subscript required in $CO_2$ Corrected.

Line 495: Should read "records" Corrected.

Line 533: Subscript required in $CO_2$ Corrected.

**References**

Barão, et al. (2015) Alkaline-extractable silicon from land to ocean: A challenge for biogenic silicon determination. *Limnology and Oceanography: Methods, 13*, 329-344. doi: 10.1002/lom3.10028

Hatton et al., (2019) Investigation of subglacial weathering under the Greenland Ice Sheet using silicon isotopes. *Geochimica et Cosmochimica Acta, 247,* 191-206. doi.org/10.1016/j.gca.2018.12.033

Hawkings et al. (2017) Ice sheets as a missing source of silica to the polar oceans. *Nature Communications, 8(1)*, 1-10. doi.org/10.1038/ncomms14198

Pickering et al. (2020**)** Using stable isotopes to disentangle marine sedimentary signals in reactive silicon pools. G*eophysical Research Letters, 47*. doi.org/10.1029/2020GL087877

---

## Author Comment (AC2)

Response to Referee#2 comments:

We thank Referee#2 for the positive and constructive comments and suggestions, and for understanding the challenge of interpreting these complex results. We have addressed all the comments and will modify our manuscript accordingly. Our responses to Referee#2 comments are written (in blue) after each comment.

**General Comments**

This is a good paper that attempts to interpret a complex data set that at first glance is not easy to understand. I commend the authors for their efforts in interpreting these records cohesively. The age models are reasonable, although in Site 17 it is more difficult to recognize where the unsupported Pb reaches supported Pb levels and the Cs peak is hard to see. Nonetheless, the authors interpretation of the likely age model is sound. The geochemistry is clear and interpreted appropriately. The authors should be commended for including detailed diatom ecological information and references to back up their interpretations. This should be universal in diatom papers but is often missing. I have some concerns and questions about the grain size data and diatom data, but overall, the paper is a great contribution to our understanding of freshwater discharge from Greenland and primary productivity.

**Specific Comments**

The figures are quite hard to read in many cases. The font is exceptionally small. In some cases, it is the equivalent of 2 or 3 points—nearly impossible to read even when zoomed way in on the pdf! In Figure 1, please increase the font size of the numbered fjords. Figures 3 and 4 are the most problematic for the reader to investigate. It's nearly impossible to read the axes because of the small font size. In addition, the authors compare productivity levels between sites, but the x-axes are different for each site. It would be helpful if the x-axes were identical between sites in Figure 3. It's possible that this will be difficult for some measurements, like BSi, but it would be helpful if possible. For Figure 4, it is inappropriate to plot diatom relative percent data with varying axes. The length of 10% on the x-axis for Sea Ice Associated species should be the same length for 10% on the x-axis for *Detonula confervacea* and for every other taxa. This is the only way to evaluate relative percent. In rare cases, it's okay to break an axis for taxa that overwhelm the assemblage, but I don't think this is an issue with this data. It needs to be clear to the reader that the assemblage is dominated by cold-water species and freshwater species are a small percentage. These axes lengths should also be the same for all three sites. In addition, the length of the axes for diatom concentration should be consistent between sites in Fig. 4. I noticed that for all of these figures, although they take up the full vertical space on the page, there is ample space for the figures to stretch horizontally to accommodate these changes in axis length. Please also increase the font size on all the axes so that they're easy to read.

We completely agree that the figures ended up too small, and that font sizes were too small. We have enlarged all figures and increased the font sizes.

We understand the principle of having similar scaling of x-axis in the figures to aid comparison, however, in this case the assemblages have many minor contributors that are important ecological indicators. By scaling the x-axes for all species, the most abundant would make it very difficult to visually discern any trends/changes in the abundance of less abundant species. Plotting different x-axes is a common practice when showing microfossil data. Therefore, we would like to maintain the x-axes in Figure 3 as originally submitted.

It's concerning that there isn't agreement between the two different grain size methods (Malvern Mastersizer and wet sieving). Shouldn't they both show increases in the coarse fraction at the same times? The authors should address this in the result section.

In this study, we used two different methods to study variations in the sediment grain sizes. These methods use different sample volumes; wet-sieving method uses larger volumes of sediment (ca. 2 grams) than the method with Malvern (ca. 0.2 grams) and is thus more sensitive to detect larger grain sizes. Malvern was specifically used to distinguish size fractions below 63 μm. Results from the two methods do not perfectly align but are complementary. We have clarified this in methods and results sections.

Please be cautious about how you interpret "productivity." In section 4.1 you discuss productivity in terms of TOC, diatoms, and BSi, which is appropriate. However, in section 4.3, you discuss productivity only in terms of BSi. It seems important that the three productivity indicators are not correlated at all sites and in some cases are anticorrelated (site 20). There should be a discussion of these differences and potentially interpretations of why they are different. This could be in Section 4.1 or a new section between 4.2 and 4.3. Please also be specific in section 4.3 that there is an increase in BSi, but not necessarily an increase in "productivity."

We have clarified throughout the manuscript which productivity proxy is discussed and added speculation why different proxies give various signals.

There is discussion about sub-glacial sediment plumes drawing nutrients from deep water to the surface. However, all these sites are at about 500 m water depth. Are the nutrients sourced from this depth (intermediate water depths) or are they somehow drawn from the deeper ocean and in over the fjord sill to upwell at the sites? I see the references here, but I'm having trouble understanding how this mechanism works. Perhaps a sentence or two more in section 4.3 would elucidate this.

We have added more text elucidating this mechanism presented in the referenced papers.

In section 4.4, I'm having a really hard time seeing the association between late summer species and freshwater discharge. I'd be more easily convinced that species like *Synedra* and benthics were responding to the freshwater increase than *T. antarctica* and *Detonula*.

At site 17 late-summer blooming cold-water species are more abundant between 1930´s and 1970´s and from mid-90´s onwards, when freshwater discharge was higher and have a positive correlation with freshwater runoff. Whereas *Synedra* spp. and benthic species do show increased abundances after the 90´s but not between 30´s and 70´s, and thus the link between the abundance of these species and freshwater runoff is not so obvious. At site 6, benthic species have a statistically significant negative correlation with inner fjord freshwater runoff.

**Technical Corrections**

Table 3: Please either include p-values or the significance level for the correlations.

We have added p-value ($p < 0.05$) in text.

Line 358: Which site are you referring to in this sentence? Site 8? This is confusing because in the first part of the sentence you refer to the Outer fjord, then say that there are frequent low-amplitude fluctuations, but I can only see one fluctuation in TOC at Site 8. Please clarify this.

Yes, here we discuss fluctuations in BSi content and diatom productions at site 8. We have made this clearer in the text.

Line 378: Please provide references for the $\delta^{13}C$ values.

We added reference to Meyers (1994).

Lines 365-371: It's striking that Site 17's BSi is similar to the other sites, but there are no diatoms at this site. It would be helpful if the authors mentioned this and perhaps addressed why this might be.

Referee#2 must refer to site 20 (and not 17) where diatoms were found to be very sparse and mainly just fragments. The BSi content at site 20 has a similar trend as BSi content at site 17, but site 20 has biogenic silica content varying between 5.3-14.3 mg/g$^{-1}$ (mean 8.0 mg/g$^{-1}$) which is lower than in sites 8, 6, and 17 where mean biogenic silica contents are 27.5, 22.2, and 12.6 mg/g$^{-1}$ (respectively). This points that site 20 has significantly lower productivity than the other sites. We have added more discussion of the missing diatom record on site 20.

Lines 385-390: It's quite easy to check and see if there is a significant amount of land-derived inorganic nitrogen. You just need to plot TOC vs. TN and check if the y-intercept is 0. If it is, then there is no clay bound nitrogen. I suggest the authors do this so they know whether or not it is present. See Schubert and Calvert (2001) for more information.

We plotted our TOC values against TN as suggested and can confirm that there is no clay bound nitrogen. We have added this to the manuscript.

Lines 416-421: Are you referring only to the inner fjord in this sentence? Please clarify. This statement doesn't hold true for Site 6.

Yes, here we discuss about diatoms in the site 17 record, and we have better clarified this in the text.

Line 528: I noticed a typo, this phrase should read, "…summer blooms become larger…" not "becomes."

We have corrected this typo. Thank you.

**References**

Schubert, C. J., & Calvert, S. E. (2001). Nitrogen and carbon isotopic composition of marine and terrestrial organic matter in Arctic Ocean sediments: implications for nutrient utilization and organic matter composition. *Deep Sea Research I* 48:789-810.